# Tamarind Seed Coat: A Catechin-Rich Source with Anti-Oxidation, Anti-Melanogenesis, Anti-Adipogenesis and Anti-Microbial Activities

**DOI:** 10.3390/molecules27165319

**Published:** 2022-08-20

**Authors:** Roongrawee Wandee, Khaetthareeya Sutthanut, Jenjira Songsri, Siriyakorn Sonsena, Ornnicha Krongyut, Patcharaporn Tippayawat, Wipawee Tukummee, Theera Rittirod

**Affiliations:** 1Department of Pharmaceutical Chemistry, Faculty of Pharmaceutical Sciences, Khon Kaen University, Khon Kaen 40002, Thailand; 2Human High Performance & Health Promotion Research Institute: HHP&HP Research Institute, Khon Kaen University, Khon Kaen 40002, Thailand; 3Bachelor of Thai Traditional Medicine, Faculty of Science, Udon Thani Rajabhat University, Udon Thani 41000, Thailand; 4Faculty of Associated Medical Sciences, Khon Kaen University, Khon Kaen 40002, Thailand; 5Department of Physiology (Neuroscience Program), Faculty of Medicine, Khon Kaen University, Khon Kaen 40002, Thailand; 6Department of Pharmaceutical Technology, Faculty of Pharmaceutical Sciences, Khon Kaen University, Khon Kaen 40002, Thailand

**Keywords:** *Tamarindus indica*, seed coat, polyphenolics, antioxidation, anti-microbial, melanogenesis, adipogenesis

## Abstract

*Tamarindus indica* L. or tamarind seed is an industrial by-product of interest to be investigated for its potential and value-added application. An ethanolic tamarind seed coat (TS) extract was prepared using the maceration technique and used to determine the phytochemical composition and bioactivities. The total phenolic and flavonoid contents were determined using colorimetric methods; moreover, chemical constituents were identified and quantified compared to the standard compounds using the HPLC-UV DAD technique. Bioactivities were investigated using various models: antioxidative activity in a DPPH assay model, anti-melanogenesis in B16 melanoma cells, anti-adipogenesis in 3T3-L1 adipocytes, and anti-microbial activity against *S. aureus*, *P. aeruginosa*, *E. coli*, and *C. albican* using agar disc diffusion and microdilution methods. The results manifested a high content of catechin as a chemical constituent and multiple beneficiary bioactivities of TS extract, including superior antioxidation to ascorbic acid and catechin, comparable anti-melanogenesis to deoxyarbutin, and significant anti-adipogenesis through inhibition of pre-adipocyte differentiation and reduction of lipid and triglyceride accumulation, and a broad spectral anti-microbial activity with a selectively high susceptibility to *S. aureus* when compared to 1% Parabens. Conclusively, TS extract has been revealed as a potential bioactive agent as well as an alternative preservative for application in food, cosmetic, and pharmaceutical product development.

## 1. Introduction

*Tamarindus indica* L. or tamarind, a member of *Leguminosae or Caesalpinaceae* family, commonly called “Ma-Kaam” in Thai, is one of Thailand’s agroeconomic products, producing as high as 140,000 tons annually [1]. Relatively, tamarind seeds are the industrial leftover by-products. All parts of this plant have been utilized along Thai-traditional remedies of cuisines, aesthetic and SPA, and herbal medicines to treat symptoms such as cold, fever, and stomach and skin disorders [2]. Despite this, scientific evidence regarding tamarind seed bioactivity has been limited; however, tamarind pulp and seeds potential for health application has been suggested by several previous reports of their phytochemical composition and biological effects: anti-inflammation, anti-microbial, anti-lipid peroxidation, inhibition of human fibroblast proliferation, and hyaluronidase and anti-tyrosinase activities [3]. Tamarind pulp is composed of various classes of phytochemicals: tannins, saponins, sesquiterpenes, alkaloids, phlobatannins, and volatile constituents―furfural, palmitic acid, oleic acid, phenylacetaldehyde, 2-furfural, and hexadecanoic acid [4,5]. Meanwhile, tamarind seeds are a rich source of polyphenolic compounds which exhibit potent antioxidation and anti-microbial activities [6,7]. The polyphenolic compounds were defined as (+)-catechin, procyanidin B2, (−)-epicatechin, procyanidin trimer, procyanidin tetramer, procyanidin pentamer, procyanidin hexamer; moreover, flavonoids such as taxifolin, apigenin, eriodictyol, luteolin, and naringenin also existed in tamarind seeds with fewer contents [8,9]. Therefore, these chemical constituents have been assumed to contribute to bioactivities found in tamarind seed extracts; this has made tamarind seeds a promising agro-industrial by-product of value for food, cosmetics, and pharmaceutical applications. Nevertheless, supporting scientific evidence that covered aspects of physicochemical composition, safety, analytical method, and beneficial health effects is acquired for rationale and sustainable utilization.

Oxidative stress is a significant fostering factor in the development of metabolic imbalance, chronic diseases, and aesthetic problems, including skin aging, hyperpigmentation, adipogenesis, obesity, etc. Oxidative stress-induced overwhelming reactive oxygen species (ROS) production in many physiological systems is caused by a genetic disorder or improper lifestyle. Consequently, accumulative damage to cell constituents and connective tissue occurs [10,11], leading to cellular function disorders and physiological imbalances, including melanocyte hyperpigmentation and adipocyte lipogenesis (adipogenesis). Melanocyte hyperpigmentation is a common sign of skin aging, which can be ameliorated by the tyrosinase inhibitors: kojic acid, vitamin C, arbutin, and deoxyarbutin—a hydroquinone derivative with equivalent efficacy and better safety profile which have been widely used in the cosmetic industry [12,13]. Obesity is developed in association with oxidative stress- and metabolic imbalance-induced hypertrophic and hyperplasic adipocytes-induced adipocyte transformation through pre-adipocyte differentiation and adipogenesis stages in the adipocyte life cycle [14,15,16].

Interestingly, the anti-obesity and anti-aging potential of polyphenolics, contained in plant extracts have been widely demonstrated in in vitro and in vivo studies [11,17,18]. The phenolics and flavonoids are members of the polyphenolics class, the well-known broad spectrum of bioactive agents. Their anti-aging and anti-obesity potential through antioxidation, anti-tyrosinase, anti-melanogenesis, and anti-adipogenesis activities as the cellular level mechanisms have been continuously convinced by many reports [19,20,21]. Therefore, the agro-industrial products and by-products containing the bioactive compounds are considered valuable resources of the Bio-Circular-Green Economic society. As a result, many agro-industrial by-products nowadays have become an interesting resource to be refined, reformed, or reused to deliver higher value derivatives or products for economic and ecological sustainability. For example, this could be found in the case of rice bran containing many bioactive agents, which has become an economic-value by-product in the health product industry after numerous supporting reports on its merits have been established. As a result, the tamarind seed coat has become an industrial by-product of interest due to its high content of phenolic compounds and good anti-microbial activity; however, the limited numbers of information about the tamarind seed coat have necessitated the exploration of its actual health benefits for applications [8]. Therefore, this study aimed to investigate the eco-industrial potential of the tamarind seed coat (TS), focusing on the characterization of phytochemical composition, chemical constituent(s), and bioactivities: antioxidative, anti-melanogenesis, anti-adipogenesis, and anti-microbial activities. The obtained results will support the introduction of tamarind seed coat as one of the agro-industrial by-products of value for applications in food, cosmetic, and pharmaceutical industries as an alternative source of bioactive agents and pharmaceutical additives to bring about economic and ecological sustainability. 

## 2. Results

### 2.1. Yields, Total Phenolic Content, Total Flavonoid Content, Antioxidant Activity, and Chemical Composition of TS Extract

The ethanolic extract of the tamarind seed coat (TS) was delivered in a dry powder form with a yield of 0.87% *w*/*w* of tamarind seed (2.01% *w*/*w* of tamarind seed coat―the outmost part with a yield of 38.51 ± 1.15% *w*/*w* of a tamarind seed). The total phenolic and flavonoid content was investigated as the phytochemical composition of TS extract. The results illustrated phenolics as the major component of TS extract rather than flavonoids, in which total phenolic content was 106.40 ± 0.69 mg Gallic acid equivalence/g extract, and total flavonoid content was 0.45 ± 0.07 mg Quercetin equivalence/g extract, respectively (Table 1). In addition, an HPLC analysis was done to identify and quantify a major chemical constituent; the results illustrated the typical chromatographic characteristic and chemical constituents of TS extract. Similar to the catechin, reference standard compound chromatogram, with a peak at RT 7.38 min and UV spectrum with λmax 209/238/278 nm (Figure 1B), a major peak at retention time (RT) 7.50 min (Figure 1(A1)) with UV spectrum (maximal absorption wavelength at λmax 209/238/279 nm) (Figure 1C) was demonstrated; this result indicated catechin as the major constituent with detected content of 429 ± 22.29 mg/g extract (approximately 43% *w*/*w* of TS extract), extrapolated from the linear equation (y = 35987x, R^2^ = 0.9984) of catechin calibration curve (Figure 1D); however, the broadening peak (RT 7.50 min) in the TS extract chromatogram, compared to the standard catechin was remarked, assuming the existence of its catechin derivatives. Interestingly, strong antioxidative effect of TS extract was exhibited with IC_50_ of 2.92 ± 0.01 µg/mL derived from a dose-response relationship with linear equation y = 17.158x (R^2^ = 0.9958) which was superior to reference standard antioxidants as Ascorbic acid (IC_50_ of 6.30 ± 0.09 µg/mL derived from a linear dose-response relationship with equation y = 7.8103x; R^2^ = 0.9962) and catechin (IC_50_ = 10.92 ± 0.14 µg/mL derived from a linear dose-response relationship with equation y = 4.5806x; R^2^ = 0.9939) (Table 1C). The obtained data have potentiated the TS extract as a rich source of catechin with a predominant antioxidant; moreover, the HPLC technique is a potential analytical method for identifying and quantifying TS extract chemical composition, which will be useful in controlling extract quality.

### 2.2. Anti-Melanogenesis of TS Extract in a B16 Melanoma Cell Line Model

The anti-melanogenesis was demonstrated by melanin content reduction after TS extract treatment in a model of forskolin-induced melanogenesis B16 melanoma cell line; this was conducted using a non-cytotoxic concentration range of samples (3.12–12.50 µg/mL with melanocyte viability >80%), compared to the controls: the negative control was forskolin-induced melanogenesis group, and the positive control was deoxyarbutin treated group, a reference whitening agent. Compared to the controls, the TS extract exhibited anti-melanogenesis (melanin content 87.63–58.60% of control) in a dose-dependent manner with a comparable degree of activity to the deoxyarbutin (melanin content 90.32–53.76% of control). The significant activity was manifested at concentrations of 6.25 and 12.5 µg/mL given 71.51 ± 5.32 and 58.60 ± 3.80% of the negative control for TS extract and melanin content 63.98 ± 0.76 and 53.76 ± 9.12% of the negative control for deoxyarbutin (Figure 2).

### 2.3. Anti-Adipogenesis of TS Extract in a 3L3-T1 Adipocyte Cell Line Model

The TS extract manifested inhibitory effects on the differentiation and adipogenesis of adipocytes. The non-cytotoxic concentration range of TS extract (0–31.25 µg/mL) with adipocyte viability of more than 80% of control (Figure 3A). At low concentrations (1.25–10 µg/mL), suppression of intracellular lipid accumulation and biotransformation of fatty acids into triglycerides, a deposited insoluble lipid form, signified the anti-adipogenesis of the TS extract. Compared to the control (differentiated adipocytes in Figure 3(C1)), the TS extract treated group showed the reduction of Oil Red O-stained lipid droplets in terms of sizes and numbers (Figure 3(C3–C6)) with a comparable character to the control (Figure 3(C2)) at a concentration of 10 µg/mL (Figure 3(C6)). Relatively, total lipid accumulation and triglyceride content were significantly diminished in a dose-dependent manner with statistical significance in a concentration range of 5 to 10 µg/mL for total lipid accumulation of 81.20 ± 2.91 and 25.82 ± 2.44% of control and 2.5 to 10 µg/mL for triglyceride accumulation of 70.09 ± 2.80, 41.12 ± 5.84, and 8.41 ± 2.35% of the control, respectively (Figure 3B).

### 2.4. Anti-Microbial Activity 

The bacterial growth inhibitory effect of the TS extract and controls (DMSO as the solvent, 1% Parabens as a reference preservative, and Ciprofloxacin and Nystatin as the standard antibiotic drugs) against *S. aureus*, *E. coli*, *P. aeruginosa*, and *C. albicans* was demonstrated in different degrees of efficacy and susceptibility to microbial types as showing the clear zone diameter differences among the sample. The TS extract and 1% Parabens fascinatingly illustrated potent anti-microbial effects with a wide spectral manner but different potency and susceptibility to each microbial type. Inhibitory effect to *S. aureus* growth of TS extract (inhibition zone diameter 14.00 ± 0.64 mm) was superior to that of 1% Parabens―as showing a larger diffusion area but with a cloudy appearance and smaller inhibition zone (diameter 9.15 ± 1.35 mm). Meanwhile, growth inhibition of the TS extract against *E. coli*, *P. aeruginosa*, and *C. albican* was inferior to 1% Parabens (Table 2). Interestingly, the TS extract exhibited high susceptibility to *S. aureus* giving lower MIC (0.03 mg/mL) and MBC (3.90 mg/mL) when compared to other tested microorganisms, *P. aeruginosa* (MIC 3.90 and MBC 15.62 mg/mL); *C. albican* (MIC 3.90 and MFC 31.25 mg/mL); and *E. coli* (MIC 7.81 and MBC 31.25 mg/mL), respectively (Table 3).

## 3. Discussion

The obtained results have revealed the TS extract as a catechin-rich source (the content as high as 429 ± 22.29 mg/g extract or approximately 42.9% *w*/*w* of extract). Catechin is a health-beneficial phenolic compound ubiquitously found in many plants but with different contents. In addition, the potent antioxidation detected in the TS extract has potentiated the promising benefits and outcomes of applications; however, there has been limited scientific evidence on phytochemical compositions and pharmacological effects of tamarind seed coat extract. Still, the reports on a high content of polyphenolics and strongly antioxidative properties of tamarind seed coat extract have drawn attention for further in-depth investigation. As the conclusive points of interest, the previous reports have evidenced different phytochemical composition, contents, and yields of tamarind seed coat extract, assuming the properties of tamarind seed coat materials, solvent types, and extraction methods as the underlining factors of the variation. Among them, methanol and ethanol were suggested as the solvents of choice to deliver a polyphenolic-rich extract with strong antioxidative activity. The methanolic tamarind seed coat extract revealed a high content of polyphenolics with a total phenolic content of 2.82 mg/g extract; they were water-soluble tannins (30%) and polymerized procyanidins (57.9%)―procyanidin tetramer (22.2), procyanidin pentamer (11.6), procyanidin hexamer (12.8), procyanidin trimer (11.3), (−)-epicatechin (9.4), procyanidin B2 (8.2), (+)-catechin (2.0) [22]. Meanwhile, an ethanolic tamarind seed coat extract contained (−)-epicatechin/catechin content of 181± 0.8 mg/kg tamarind seed coat [9]. The composition of potent antioxidants in tamarind seed coat extract as 2-hydroxy-3’, 4’-dihydroxyacetophenone, methyl 3,4-dihydroxybenzoate, and 3,4-dihydroxyphenyl acetate was also illustrated [23]. In contrast, the ethyl acetate solvent-derived tamarind seed coat extract contained a less phenolic content (85.6 ± 0.9 mg catechin equivalence/g extract; 8.56%), but with preventive and treatment activity on melanocyte-stimulating hormone (MHS)-induced melanogenesis in B16-F1 melanoma cells [24]. In harmony, the obtained result displayed anti-melanogenesis effect of TS extract with a comparable degree to the deoxyarbutin—a glycoside derivative of hydroquinone with a strong tyrosinase inhibitory effect. The contribution of catechin, the major constituent (42.9% *w*/*w* of extract) of TS extract, was presumed through tyrosinase inhibition due to the ability of hydrogen bonding to the active site of the enzyme from its hydroxyl groups [25]; moreover, catechin contribution has been supported by a report on its potent antioxidation and tyrosinase inhibition in B16 melanoma cells [26]. Catechin is a member of phenolic compounds, also known as the flavan-3-ols or flavanols―its core structure with an aromatic ring is attached to five hydroxyl groups, which mainly exist in chocolate, grapes, green tea, and wine. Notably, green tea leaf is the richest source of polyphenolics in approximately 10–30% (*w*/*w* by dry weight) of catechin compounds―catechin, epicatechin (EC), epigallocate catechin (ECG), and epigallocatechin gallate (EGCG), in which EGCG is the major component [27,28]. As a result, green tea is widely known as a health-promoting natural product. Interestingly, our finding demonstrated a predominant content of catechin as high as 42.9% *w*/*w* of extract in TS extract or 429 ± 22.29 mg/g extract; this has potentiated tamarind seed coat as an alternative catechin-rich source of interest. In addition, catechin has been suggested as a potential chemical marker to be used in tamarind seed coat extract quality control; however, further investigation to identify and clarify the actual existing form of catechin and derivatives should be considered.

The existence of catechins polymerized forms and derivatives in the TS extract has been presumed; this has been suggested from the demonstration of a potent antioxidative activity of the TS extract over the catechin standard compound (Table 1C), which has agreed with previous findings of Liang, et al. [29], Procházková, et al. [30]; this assumption has been potentiated by evidence of the advantageous properties of oligomeric catechin as proanthocyanidins compared to monomeric catechin. Polymerization of catechins commonly occurs via plant enzymatic reaction to form oligomeric catechin, which possesses thermal stability and efficacy. These include antioxidation associated with poly-hydroxyl groups in their molecules [29,30], superoxide scavenging capacity, inhibition of the xanthine oxidase (XO) enzyme, and peroxidation of low-density lipoprotein (LDL) in a correlation with their molecular weights [31,32]; moreover, a polymerization into oligomers of catechin could intensify anti-microbial efficacy against Gram-positive and Gram-negative bacteria through the destruction of bacterial cells [33]. By contrast, concerning issues of monomeric forms of phenolics and flavonoids on their sensitivity to light, heat, and alkaline conditions, poor bioavailability; rapid metabolism; and poor membrane permeability have been evidenced, which relative may result in a burden on industrial feasibility and application [34,35]. Therefore, information on the stability and the existing forms of catechin in the TS extract are crucial to be elucidated.

The health benefit of TS extract is potentiated by the evidence of catechin biological activities, which covers a wide range of physiological systems, including potent antioxidative, anti-obesity, anti-diabetes, and anti-microbial activity. The beneficiary application of TS extract to obesity and metabolic syndromes is a promising expectation related to its antioxidative and anti-adipogenesis potential. The oxidative stress-induced obesity and type 2 diabetes development through molecular and cellular disorders is associated with stimulating cascades of inflammation and adipocyte dysfunction in adipose tissue. These result in adipogenesis and intracellular lipid accumulation in liquid and insoluble fat (triglyceride) droplet forms, anticipating by promoting gene expression of adipogenic genes―the transcription factors of CCAAT/enhancer-binding protein (C/EBPα) and peroxisome proliferator-activated receptor-gamma (PPARγ) [36,37,38]; moreover, the antioxidative-related anti-adipogenesis effects of catechin induced S phase arrest during adipogenic differentiation and anti-adipogenesis in fat cells were fascinatingly revealed [36]. As a piece of concrete evidence, the anti-obesity property of catechin was successfully proven in clinical trials. As supporting reports, preventive potential in obesity and metabolic syndrome of polymerized catechins in green tea anticipated by its anti-obesity activity and improvement of insulin resistance, vascular function, and cardiac hypertrophy was signified in the green tea products [17]. A similar outcome was also denoted in polyphenolic-rich wine resulting in obesity improvement and blood glucose control in patients with type 2 diabetes [18].

The anti-microbial effect of TS extract in a broad spectral character but different susceptibility to each studied microorganisms, in a priority of *S. aureus*, *P. aeruginosa*, *C. albican,* and *E. coli* was demonstrated. As supporting evidence, the anti-bacterial activity of ethanolic tamarind seed coat extract against both gram-positive and gram-negative bacteria was reported: *Bacillus subtilis*, *Bacillus megaterium*, *Staphylococcus aureus, Saarcina lutea, Shigella dysentriae*, *Escherichia coli*, *Salmonella typhi* and *Salmonella paratyphi* was illustrated in a previous report with an unknown mechanism or active compound [39]. Related to the substantial content of catechin and previous evidence on its anti-microbial activity, catechin has been presumed to share anti-microbial activity found in TS extract. Catechin is a potential anti-bacterial agent involving multiple mechanisms, leading to damage of bacterial cell walls. The mechanism of action has been ascribed to an interaction between catechin and the hydrophobic domains of bacterial lipid bilayers [40] which the number of hydroxyl groups on the B-ring, the presence of the galloyl moiety, and the stereochemical structure of catechin(s) govern their affinity for lipid bilayers [41]. In addition, the influencing capacity of catechin to reduce β-lactam resistance correlates closely with their degree of penetration into the phospholipid barrier [42]. In addition, multi-drug resistance deactivating is a possible mechanism related to dimeric catechins (proanthocyanidins) and its photolytic reaction product, a superoxide anion radical (O∙^−^^2^), which suppressed growth and deactivated multi-drug-resistance of *Acinetobacter baumannii* bacterium [43,44].

The high susceptibility to *S. aureus* of the TS extract was interestingly presented by a potent inhibition expressed by a larger clear zone diameter and a low MIC (less than 10 mg/mL); this implies the particular merits of TS extract for dermatological applications. Interestingly, the magnificent inhibitory effect on *S. aureus* growth (inhibition zone diameter 14.00 ± 0.64 mm) superior to that of 1% Parabens―as showing more diffusion area but with a cloudy appearance and smaller inhibition zone (diameter 9.15 ± 1.35 mm) was displayed. Potentially, the TS extract can become an alternative preservative agent for skin products with fruitful outcomes from its multiple biofunctions. Nevertheless, additional investigation of its stability and safety should be considered. Opposite to Parabens, the health benefits of TS extract rather than risks or adverse effects are potentially expected in long-term utility. Parabens are the most well-known group of chemical preservatives, alcohols and derivatives containing hydroxyl groups (-OH) and alkyl chain in the molecules. Anti-microbial activity increases as the carbon number of the alkyl chain in the molecule; however, it is active only in the water phase—the higher pH values, the more paraben dissociation, and the higher anti-microbial activity provided [45]. Critically, the concerns related to parabens’ usage have been raised due to reports on toxicity and risks for health and the environment with a great extent from mild to severe levels. Parabens-induced adverse effects have included allergic dermatitis, breast cancer development associated with estrogen-like structure of parabens, DNA damage by 8-Oxo-2′-deoxyguanosine―a metabolite of methylparaben transformation, and skin esterase metabolism [46,47,48]. Therefore, the discovery of potential natural extracts is motivated. The overall results from this study will fortify a promising future for TS extract to become an alternative source of bioactive ingredients which function in mutual properties of health beneficiary impact and product preservation.

## 4. Materials and Methods

### 4.1. Reagents and Instruments 

Butanol, phenol red-free dulbecco’s modified eagle’s medium (DMEM), fetal calf serum (FCS), fetal bovine serum (FBS), and penicillin–streptomycin mixture were purchased from Invitrogen (Carlsbad, CA, USA). Dexamethasone (DEX), 3-isobutyl-1-methylxanthine (IBMX), insulin, Oil Red O solution, triacylglycerol assay kit, free glycerol reagent, 3-(4,5-dimethyl-2-thiazolyl)-2,5-diphenyl-2H-tetrazolium bromide (MTT), dimethyl sulphoxide (DMSO), 2,4,6-Tris(2-pyridyl)-s-triazine (TPTZ), ethylenediamine tetraacetic acid (EDTA), polyethylene glycol tert-octylphenyl ether (Triton^TM^ X-100), tris(hydroxymethyl)aminomethane, potassium chloride (KCl), acetic acid, sodium acetate, sodium lauryl sulfate (SLS), isopropanol, ethanol, methanol, trypsin-EDTA solution 0.25%, forskolin, ascorbic acid, gallic acid, quercetin, catechin, and sodium carbonate, Sodium chloride, Ciprofloxacin, and Nystatin were purchased from Sigma (St. Louis, MO, USA). Concentrated parabens (mixture of 10% methyl paraben and 2% propyl paraben in propylene glycol) was purchased from Hong Huat Company (Bangkok, Thailand). B16 melanoma and 3T3-L1 adipocyte cell lines were purchased from Lonza Bioscience (New Hampshire, MA, USA). Mueller Hinton Agar (MHA), Mueller Hinton Broth (MHB), and Brain Heart Infusion Broth (BHI) from HI-MEDIA^®^ (Mumbai, India). *Staphylococcus aureus* (ATCC 2921). *Escherichia coli* (ATCC 25922). *Pseudomonas aeruginosa* ATCC (27803). *Candida albican* obtained from Faculty of Associated Medical Sciences, Khon Kaen University. All other analytical grade chemicals were purchased from Ajax Finechem (Auckland, New Zealand). The instruments used were an inverted microscope (Axio Vert.A1 FL LED, ZEISS^®^, Jena, Germany), a microplate reader 96-well plate (VarioskanTM Flash Multimode Reader, Thermo Scientific^®^, MA, USA), a UV spectrophotometer (UV-1700, Shimazu, Japan), a centrifugation machine (Kubota, Tokyo, Japan), a High-performance liquid chromatography (HPLC) system (Thermo Scientific, Massachusetts, USA), a hot air oven (France Etuves, France), Whatman paper No.1 (N-1000, Tokyo Rikakikai Co. Ltd., Japan), a rotary evaporator (Buchi, Flawil, Switzerland) and a lyophilizer (Scanvac, Lynge, Denmark), Sircol^®^ Kit (Biocolor Ltd., Carrickfergus, UK). *T. indica* (Ma-Kaam in Thai; TS) seeds; oval square-shaped dark brown color seeds derived from the ripen tamarind fruits and kept at room temperature for a 1-mon period, were purchased from Khon Kaen province of Thailand.

### 4.2. Preparation of Tamarind Seed Coat Extract

The tamarind seeds were washed with distilled water, dried at an ambient condition, mixed with analytical grade ethanol as the solvent in a ratio of 1:5 (*w*/*v*), sonicated at 50 ± 2 °C for 20 min, and followed with 24 h maceration. The filtrate was collected using a filter membrane (Whatman^®^ No.1) and consequently concentrated using rotary evaporation and lyophilization. The yielded extract was calculated, expressed in *w*/*w* of the dry weight of origin(s), and kept at –20 °C for further experiment.

### 4.3. Determination of Antioxidant Activity, Phenolic Content, and Flavonoid Content of TS Extract

#### 4.3.1. Total Phenolic Content

The total phenolic content of TS extract was determined using the Folin-Ciocalteu method [49] by extrapolating from a calibration curve of Gallic acid, a reference phenolic compound (Table 1B). First, the calibration curve was established by plotting Gallic acid at various concentrations (0 to 100 µg/mL) and their corresponding absorbencies to obtain a linear equation (y = ax) and regression coefficient (R^2^) (Figure 2). Then, aqueous solutions of all components were prepared and used to constitute the reaction mixture. The mixture comprised of 50 µL of sample solution (various concentrations of standard gallic acid solutions or 250 µg/mL TS extract solution), 25 µL of 1 N Folin-Ciocalteu reagent, and 125 µL of 20% *w*/*v* sodium carbonate and followed with a 40-min incubation. Then, absorbance at 700 nm wavelength was measured using a spectrophotometer. Then, the total phenolic content was calculated by following the equation and presented as mg of gallic acid equivalent /g of extract.
(1)$$\mathrm{Total}\;\mathrm{phenolic}\;\mathrm{content}=\frac{{\mathrm{Abs}\;}_{\mathrm{sample}\;}-{\mathrm{Abs}\;}_{\mathrm{blank}\;}}{\;\mathrm{Slope}\;\times \;\mathrm{Amount}{\;}_{\mathrm{sample}}\;(g)\;}$$
where Abs _sample_ is the absorbance of sample solution, Abs _blank_ is the absorbance of blank, Slope is the “a” derived from the linear equation (y = ax) of the gallic acid calibration curve, and Amount _sample_ is the amount of sample (g).

#### 4.3.2. Total Flavonoid Content

Using a modified method from Aryal, Baniya, Danekhu, Kunwar, Gurung and Koirala [49], total flavonoid contents in the extracts were determined. First, a calibration curve plotting between Quercetin concentrations (0 to 100 µg/mL) and their corresponding absorbencies was established using 0.1 mg/mL methanolic quercetin solution as the stock solution. Then, using methanolic solutions of all components, a reaction mixture was constituted, comprising of 50 µL of (various concentrations of Quercetin solutions or 1000 µg/mL TS extract solution) and 50 µL of 2% AlCl_3_ solution and followed with a 40-min incubation. Then, absorbance at 432 nm wavelength using a spectrophotometer. Using the linear equation (y = ax) with regression coefficient (R^2^ ≅ 1) of the Quercetin calibration curve (Table 1A), total flavonoid content was calculated by following the equation and presented as mg of Quercetin equivalent/g of extract.
(2)$$\mathrm{Total}\;\mathrm{flavonoid}\;\mathrm{content}=\frac{\mathrm{Abs}{\;}_{\mathrm{sample}}\;-\;\mathrm{Abs}{\;}_{\mathrm{blank}}\;}{\;\mathrm{Slope}\;\times \;\mathrm{Amount}{\;}_{\mathrm{sample}}\;(g)\;}$$
where Abs _sample_ is the absorbance of sample solution, Abs _blank_ is the absorbance of blank, Slope is the “a” derived from the linear equation (y = ax) of the quercetin calibration curve, and Amount _sample_ is the amount of sample (g)

#### 4.3.3. Antioxidant Activity

Antioxidant activity of TS extract was evaluated according to the DPPH (1,1-diphenyl-2-picrylhydrazyl) scavenging method compared to reference standard antioxidants, ascorbic acid (Vitamin C) and catechin. Working solutions of the samples (or reference standard antioxidant) were separately prepared in methanol to achieve an optimal final concentration range; 62.5–2000 µg/mL for TS extract and 1.25–20 µg/mL for ascorbic acid and catechin. Then, the reaction mixture was constituted in a 96-well plate, comprising 150 µL of sample solution and 50 µL of 0.2 mM DPPH (2, 2 diphenyl-1-picryl-hydrazyl) solution. Following a 15-min incubation at an ambient condition, optical absorbance at 570 nm wavelength was measured using a spectrophotometer. The percentage of inhibition compared to the control (% inhibition) was calculated following the equation.
(3)$$\%\;\mathrm{inhibition}=\frac{\mathrm{Abs}{\;}_{\mathrm{control}}\;-\;\mathrm{Abs}{\;}_{\mathrm{sample}}}{\mathrm{Abs}{\;}_{\mathrm{control}}}\;\times \;100$$
where Abs _control_ is the absorbance of methanolic DPPH solution and Abs _sample_ is the absorbance of TS extract (or standard compounds) treated group. An effective concentration of 50% antioxidation capacity (IC_50_) of each sample was determined from a linear concentration-response curve, plotting concentration versus %inhibition before extrapolating for IC_50_ from the linear regression equation (with regression coefficient (R^2^)) (Table 1C).

### 4.4. Chemical Constituent Analysis Using High-Performance Liquid Chromatography

The methanolic solution of the TS extract sample at a concentration of 2.5 mg/mL and standard compound (catechin) at various concentrations (0–400 µg/mL) were separately prepared and used for HPLC analysis. The analysis was done in an HPLC system of a Surveyor PDA Plus UV detector at wavelengths of 254, 280, and 360 nm, equipped with a stationary phase of C18 column (ZORBAX SB-CN column, particle size 3.5 µm, dimension 4.6 × 150 mm) and a programmatic gradient mobile phase system. The gradient program of a mixture of methanol (solution A) and 2% *v*/*v* formic acid in deionized water (solution B) at a flow rate of 1 mL/min comprised of increasing solution A from 0 to 100% within 10 min (0.01–10min), then hold 100% solution A for 10 min (10.01–20min), increasing solution B to 100% within 5 min (20.01–25min) and maintain 100% solution B for 5 min (25.01–30min), respectively. In each run, a 5 µL of the sample (or standard compound) solution was injected into the HPLC system to undergo a separation and detection process of chemical constituents and finally deliver a chromatogram containing the chemical constituent peak(s) with corresponding retention times (RTs), peak area, and UV spectrum having specific maximal absorption wavelength (λmax; nm). Compared to catechin—the reference standard compound, the prominent peak of the TS extract chromatogram was identified and confirmed by a spike technique. The catechin spiked TS extract mixture solutions, 10 and 500 µg/mL final concentrations of catechin in 1 mg/mL TS extract solution, were prepared and injected into the HPLC system; moreover, the catechin content (mg/g extract) was quantified by extrapolating from a catechin calibration curve—the linear plot between average peak areas (y-axis) and corresponding concentrations (x-axis) with a linear equation (y = ax + b) and regression (R^2^).

### 4.5. Anti-Melanogenesis Activity in a B16 Melanoma Cell Model

Using a modified method described by Jampa, et al. [50], the 90% confluence of sub-passage#5 B16 culture in 10% FBS-supplemented DMEM culture media was prepared before being seeded into each well of a 96-well plate with a density of 20,000 cells/well and followed with 24-h incubation at maintenance conditions −37 °C temperature with 5% CO_2_ atmosphere; this was used for the experiment. 

Cell viability test using MTT assay was conducted by treating the cells with various concentrations of TS extract solution or deoxyarbutin (a standard compound) and incubated under maintenance conditions for 48 h. Then, the supernatant was replaced with 50 µL of MTT solution (0.5 mg/mL), incubated under maintenance conditions for 4 h, gently removed before adding 50 µL dimethyl sulfoxide (DMSO), followed by 1-min mixing, and measured the absorbance at 570 nm wavelength using a microplate reader. Compared to the control (untreated group), the cell viability was calculated and expressed as the % viability of the control using the following equation: % viability = (Absorbance of sample/Absorbance of the control) × 100. The non-cytotoxic concentration range was defined by the 80–100% cell viability (of the control).

Having deoxyarbutin as the reference standard compound, the anti-melanogenesis activity was determined by treating the cells with various concentrations of TS extract solution (or deoxyarbutin)—prepared in 10% (*v*/*v*) FBS and 50 µM forskolin-supplemented DMEM culture medium, followed by 48-h incubation under maintenance conditions. The forskolin-treated group cultured in 10% FBS-supplemented DMEM was used as the (positive) control. After 48 h incubation, the supernatant was removed, 100 µL 1 N NaOH was added to each well, incubated at 70 °C in a hot air oven for 1 h, and followed with a 2-min sonication. Then, the absorbance at a 405 nm wavelength was measured using a microplate reader. Compared to the control, % melanin content was calculated using the following equation and expressed as %Melanin content (of control).
%Melanin content (of control) = (A _sample_/A _control_) × 100(4)
where A _control_ is the absorbance of the control and A _sample_ is the absorbance of the tested sample.

### 4.6. Anti-Differentiation and Anti-Adipogenesis Activity in a 3T3-L1 Adipocyte Model

#### 4.6.1. Pre-Adipocyte Maintenance and Differentiation

Using the method of Chaiittianan, et al. [51], the 7th sub-passaged 3T3-L1 murine pre-adipocytes culture was maintained in the pre-adipocyte medium (PM); 10% *v*/*v* fetal calf serum (FCS) supplemented Dulbecco’s modified eagle medium (DMEM) containing 1% penicillin-streptomycin until reached 90% confluence. The differentiated adipocytes were prepared by seeding pre-adipocytes into 24-well plates with a density of 24,000 cells/well, maintained at 37 °C in a humidified atmosphere of 5% CO_2_ until 80% confluence, and treated with differentiation media in a 6-day period each with a priority of differentiation medium I (DMEM supplemented with 10% FCS, 1 µM dexamethasone; DEX, 500 µM 3-isobutyl-1-methylxanthine; IBMX, and 10 µg/mL insulin) and differentiation medium II (DMEM supplemented with 10% FCS and 10 µg/mL insulin) and medium III (DMEM, 10% Fetal bovine serum; FBS). The differentiated adipocytes were used for further experiments.

#### 4.6.2. Effect on Pre-Adipocyte Viability by Using MTT Assay

The cytotoxic effect and non-cytotoxicity concentration range of TS extract for the further experiment were defined. Pre-adipocytes (4000 cells/well) maintained in 96-well plates were treated with various concentrations of TS extract solution prepared in PM culture medium, incubated under 37 °C in a humidified atmosphere of 5% CO_2_ for 48 h, and subsequently added 50 µL of MTT solution (5 mg/mL in phosphate-buffer saline) to each well and followed by a 4 h incubation at 37 °C temperature, removed the supernatant, added 100 µL of DMSO and followed by 15-min incubation at room temperature. Then, the absorbance was measured at 560 nm using the microplate spectrophotometer. The %viability (of control) of each sample was calculated compared to the control.

#### 4.6.3. Anti-Adipogenesis by Using Oil Red O Staining and Triglyceride Assay

Various concentrations of TS extract solution were prepared in differentiation medium type I–III. Then, the pre-adipocytes were treated with sample solutions following the condition mentioned in Section 4.6.1: Pre-adipocyte maintenance and differentiation. The cells maintained in the differentiated media (without TS extract) were used as the control. Detection of cell differentiation and intracellular lipid accumulation of adipocytes were conducted by using Oil Red O staining and triglyceride assay, respectively.

Oil red O staining method, the differentiated adipocytes were fixed with 10% (*v*/*v*) formalin for 30 min, rinsed with PBS (pH 7.4), stained with freshly prepared 0.5% (*w*/*v*) Oil Red O solution at 37 °C for 1 h, and the stained cells were photographed using an inverted microscope. Then, the intracellular retained dye was extracted with 100% isopropanol and measured optical absorbance at 510 nm using a microplate spectrophotometer to quantify the accumulative lipid content. Compared to the control, the inhibitory effect of each sample reflected by accumulative lipid content was calculated following the equation and expressed as % lipid accumulation (of control).
(5)$$\%\mathrm{lipid}\;\mathrm{accumulation}\;(\mathrm{of}\;\mathrm{control})=\frac{\mathrm{Abs}{\;}_{\mathrm{sample}}\;}{\mathrm{Abs}{\;}_{\mathrm{control}}}\;\times \;100$$
where Abs _sample_ is the absorbance of sample and Abs _control_ is the absorbance of control.

The effect on triglyceride accumulation was evaluated. The treated differentiated adipocytes were washed twice with ice-cold PBS (pH 7.4), scraped in 300 µL of lysis buffer (0.15 M NaCl, 10 mM EDTA, 0.1% Triton^TM^ X-100, 50 mM Tris buffer pH 7.4), and followed by 10-min sonication. The lysed cell mixture was centrifuged at 12,000× *g* for 10 min under 4 °C temperature, to collect the supernatant for quantitative analysis of the triglyceride (triacylglycerol) content by using a serum triacylglycerol determination kit. The optical absorbance at 540 nm wavelength was measured using the microplate spectrophotometer. Compared to the control, the relative triglyceride content was expressed as triglyceride accumulation (% of control) following the equation.
(6)$$\mathrm{Triglyceride}\;\mathrm{accumulation}\;(\%\;\mathrm{of}\;\mathrm{control})=\frac{\mathrm{Abs}{\;}_{\mathrm{sample}}\;}{\mathrm{Abs}{\;}_{\mathrm{control}}}\;\times \;100$$
where Abs _sample_ is the absorbance of sample and Abs _control_ is the absorbance of control.

### 4.7. Anti-Microbial Activity Using Agar Disc Diffusion Assay and Drop Plate Technique

#### 4.7.1. Preparation of Medium

Mueller Hinton Agar (MHA) plate: dissolved 38 g of MHA powder in 1 L of distilled water, autoclave at 121 °C for 20 min, aliquoted sterile MHA agar mixture into Petri dishes of 21 mL each, dried in a hot air oven at 60 °C, and stored in a refrigerator until used; Mueller Hinton Broth (MHB) and Brain Heart Infusion (BHI) Broth were separately prepared: dissolved 20 g of MHB powder (or 37 g of BHI powder) in 1 L of distilled water, autoclave at 121 °C for 20 min, and stored in a refrigerator until used.

#### 4.7.2. Preparation of Microorganism Suspension

*Staphylococcus aureus* ATCC 29213, *Escherichia coli* ATCC 25922, *Pseudomonas aeruginosa* ATCC 27803, and *Candida albican* (from Faculty of Associated Medical Sciences, Khon Kaen University, Khon Kaen, Thailand) were used to prepare the culture stock for the experiment. The single colony of each bacterium, obtained from the MHA agar steak-plate technique followed with 12 h incubation at 37 °C, was inoculated into a 3-mL of proper broth medium—MHB for *S. aureus*, *E. coli,* and *P. aeruginosa*) or BHI for *C. albican*, for 12 h under 37 °C. Then, the culture was collected and rinsed with 0.9% *w*/*v* sterile sodium chloride solution, followed by centrifugation at a speed of 9000 rpm for 5 min under a temperature of 4 °C. Finally, the rinsed pellets were resuspended in 0.9% *w*/*v* sterile sodium chloride solution to achieve a density of 1 × 10^8^ CFU/mL bacterial suspension for *S. aureus*, *E. coli,* and *P. aeruginosa*, and 1 × 10^6^ CFU/mL for *C. albican*. The prepared microorganism suspension was used in the experiments.

#### 4.7.3. Screening of Anti-Microbial Activity by Using Agar Disc Diffusion Assay

TS extract solution dissolved in Dimethyl sulfoxide (DMSO) was prepared at a concentration of 250 mg/mL and used for the anti-microbial experiment. Screening for anti-microbial activity against *S. aureus*, *E. coli*, *P. aeruginosa,* and *C. albican* of the TS extract was performed using a modified agar disc diffusion test described in Nascente Pda, et al. [52], Wikler [53]. The diffusion discs were placed onto the prepared MHA gar plate―superficially spread over with microbial suspension before applying 10 µL of each sample onto a disc. There was TS extract solution or the controls: DMSO as a negative control and 1% Parabens and ciprofloxacin (or nystatin) as positive controls. Then, plates were incubated overnight at 37 °C and measured inhibition zone diameter using a Vernier caliper. The independent experiment was done in duplicate. Consideration of potential, the inhibition zone of TS extract was compared to 1% Parabens as a reference pharmaceutical preservative and Ciprofloxacin or Nystatin as a standard anti-bacterial and anti-fungal drug, respectively.

#### 4.7.4. Determination of Minimal Inhibitory Concentration, Minimal Bactericidal Concentration, and Minimum Fungicidal Concentration 

Using a modified microdilution method ascribed by Nascente Pda, Meinerz, de Faria, Schuch, Meireles and de Mello [52], minimal inhibitory concentration (MIC), minimal bactericidal concentration (MBC), and minimum fungicidal concentration (MFC) were defined. Briefly, a serial two-fold (1:2) dilution of TS extract solution was prepared using a proper broth media (MHB for bacterial and BHI for fungal culture). The mixture of 50 µL sample solution (to achieve the final concentration range of 0.06–125 mg/mL) and 50 µL of inoculum microorganism suspension was constituted in a 96 well-plate. Then, the mixture in the 96 well-plate was incubated at 37 °C overnight and subsequently subjected to the determination of viable colony numbers using the drop plate technique. In addition, the experimental condition was verified by the control group: a mixture of microbial and the medium as a negative control and 1% Parabens and Ciprofloxacin (or Nystatin) as positive controls (the experimental layout in Appendix A).

Drop plate technique, a 10 µL of each sample mixture was carefully dropped onto the MHA plate in duplication. After overnight incubation at 37 °C, colony formation was observed and considered to indicate the MIC, MBC, and MFC. The colony numbers of the treated group are compared to that of the control (untreated group; microbial + medium). The MIC was indicated at the lowest concentration, resulting in colony density and size reduction as evidenced by the bacterial growth inhibitory effect. In contrast, the MBC or MFC was accounted for at the lowest concentration, resulting in no bacterial or fungal colony formation.

### 4.8. Statistics Analysis

All experiments were done in triplication, except microbiological tests in duplication. The results were expressed as mean ± standard deviation. The statistical analysis using SPSS version 19 statistical software program by One-Way ANOVA analysis of variance and multiple comparisons were conducted. The significance was taken at *p*-value less than 0.05 (*p*-value < 0.05).

## 5. Conclusions

Tamarind seed coat (TS) extract was revealed as a catechin-rich bioactive agent. These included superior antioxidation to ascorbic acid and catechin, comparable anti-melanogenesis to deoxyarbutin, significant anti-adipogenesis through inhibiting differentiation and reducing lipid and triglyceride accumulation, and a broad spectral anti-microbial activity with selectively high susceptibility to *S. aureus*. Conclusively, the results have potentiated tamarind seed coat to become an agro-industrial by-product of value to deliver bioactive agents and pharmaceutical additives for food, cosmetic, and pharmaceutical applications; however, skin sensitivity, irritation, toxicity, and pharmacological effects at high concentration usage should be considered for maximal benefits and sustainability of utilization.

## Figures and Tables

**Figure 1 molecules-27-05319-f001:**
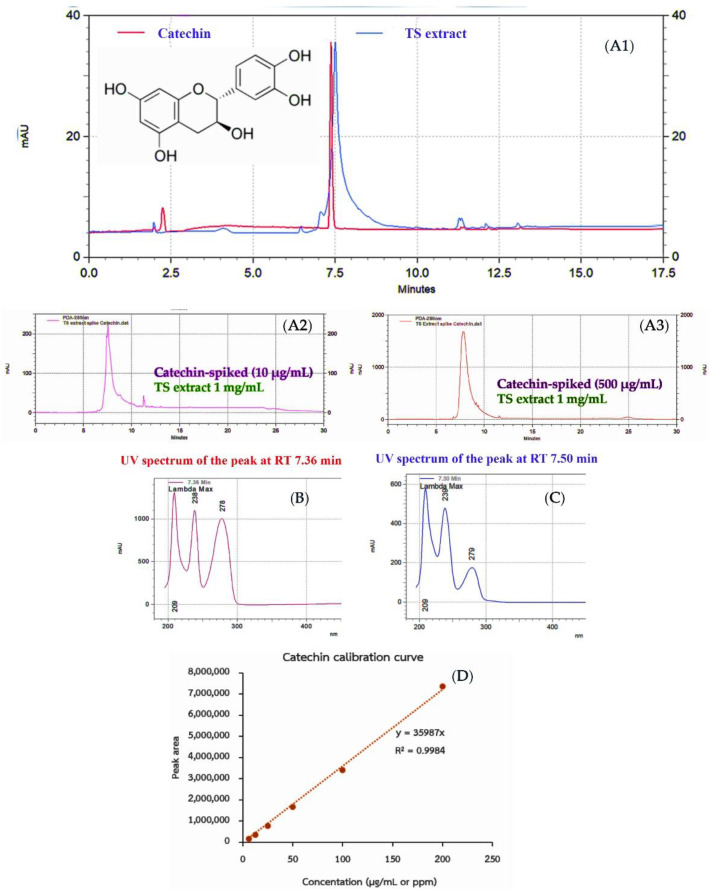
The HPLC chromatogram of the tamarind seed coat (TS) extract solution (concentration of 2.5 mg/mL) (**A1**, blue chromatogram) contained a prominent peak at retention time (RT) of 7.50 min with a specific pattern of UV spectrum having a maximal absorption wavelength (λmax) at 209/239/279 nm (**C**), which was similar to the reference standard catechin―a peak at RT 7.36 min (**A1**, red chromatogram) with UV spectrum having λmax at 209/238/278 nm (**B**). The catechin-spiked TS extract mixture chromatograms demonstrated the co-elution of catechin and the prominent peak in TS extract chromatograms (**A2**,**A3**). Thus, Catechin was identified as a major constituent of TS extract with content as high as 429 ± 22.29 mg/g extract, extrapolating from the catechin calibration curve (**D**).

**Figure 2 molecules-27-05319-f002:**
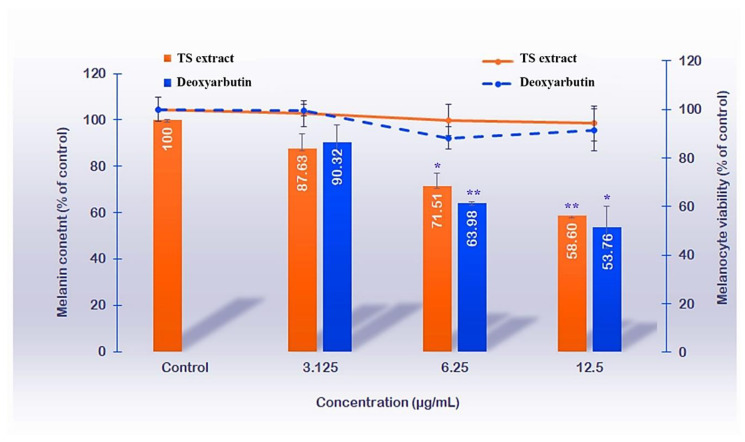
The anti-melanogenesis effects of the tamarind seed coat (TS) extract and deoxyarbutin were displayed by melanin content reduction (bar) at non-cytotoxic concentrations—given >80% cell viability of control (line). The significant effect was detected at 6.25 and 12.5 μg/mL TS extract (or deoxyarbutin) concentration, compared to the control (* *p*-value < 0.05; **, *p*-value < 0.01).

**Figure 3 molecules-27-05319-f003:**
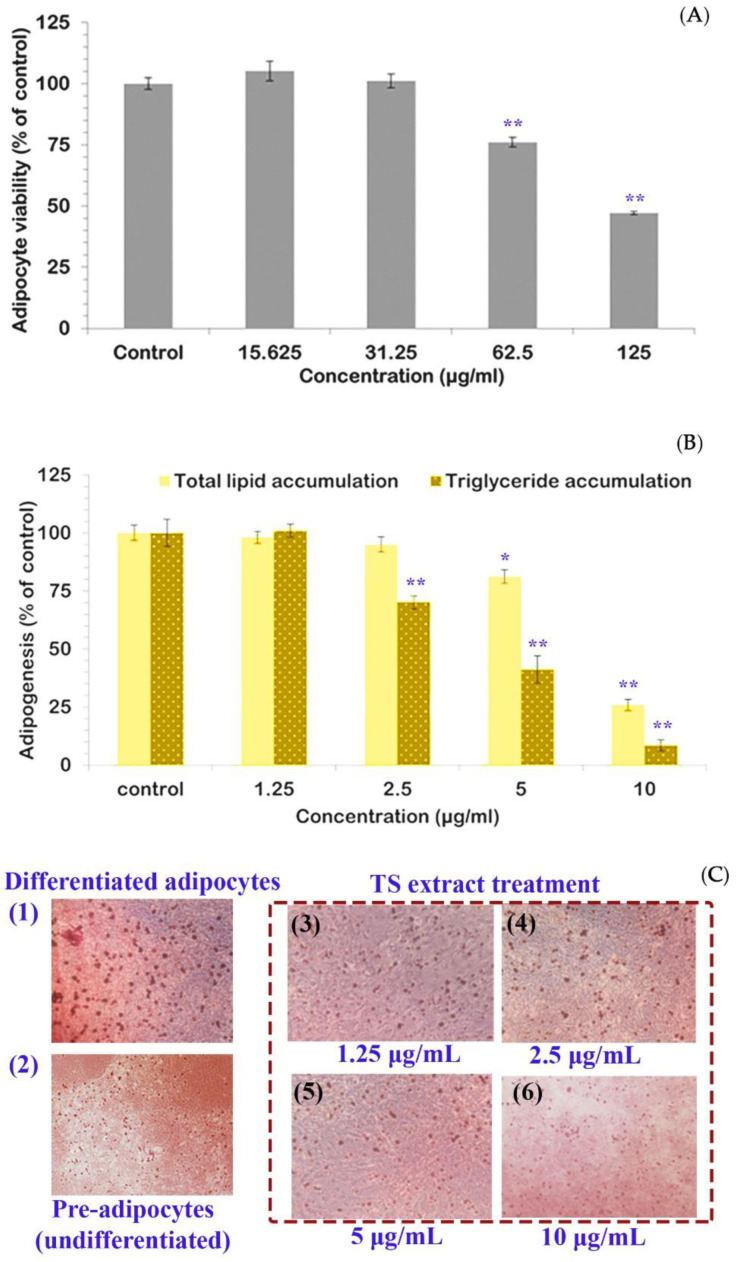
Tamarind seed coat (TS) extract manifested anti-adipogenesis in a dose-dependent manner. The illustrated results were achieved at a low concentration range (1.25–10 μg/mL), given adipocyte viability >80% of control (**A**). The reduction of Oil Red O-stained cell numbers in the treatment group when the TS extract concentration increased was demonstrated under an inverted microscope (magnification 20x) (**C**) with a comparable effect to the undifferentiated group (2) at TS extract concentration of 10 μg/mL (6). In addition, reduction of lipid accumulation was manifested with statistical significance at concentrations of 5 and 10 μg/mL, compared to the control (* *p*-value < 0.05). Relatively, TS extract at concentrations greater than 2.5 μg/mL significantly reduced the accumulation of triglyceride when compared to the control (** *p*-value < 0.01) (**B**).

**Table 1 molecules-27-05319-t001:** Tamarind seed 100 g composed of seed coat 38.51 g yielded 0.77 g of the tamarind seed coat (TS) extract (2.01 %*w*/*w*) which predominantly contained phenolic compound (106.40 ± 0.69 mg/g) rather than flavonoids (0.45 ± 0.07 mg/g), the contents were extrapolating from the standard curves of Gallic acid (Figure **A**) and Quercetin (Figure **B**), respectively; moreover, the superior antioxidant activity of TS extract (IC_50_ 2.92 ± 0.01 µg/mL) to the standard antioxidants, ascorbic acid (IC_50_ 6.30 ± 0.09 µg/mL) and catechin (IC_50_ 10.92 ± 0.14 µg/mL) was demonstrated, expressed by 50% inhibitory concentration (IC_50_) derived from the concentration-response curves from DPPH assay (Figure **C**).

Yield of seed coat	38.51 ± 1.15% *w*/*w* of tamarind seed
Yield of seed coat extract	0.87% *w*/*w* of tamarind seed
	2.01% *w*/*w* of seed coat
Total phenolic content	106.40 ± 0.69 mg gallic acid equivalence/g extract
Total flavonoid content	0.45 ± 0.07 mg quercetin equivalence/g extract
Antioxidant activity	IC_50_ 2.92 ± 0.01 µg/mL ^Ψ^

^Ψ^ Compared to ascorbic acid IC_50_ 6.30 ± 0.09 µg/mL and catechin IC_50_ 10.92 ± 0.14 µg/mL.

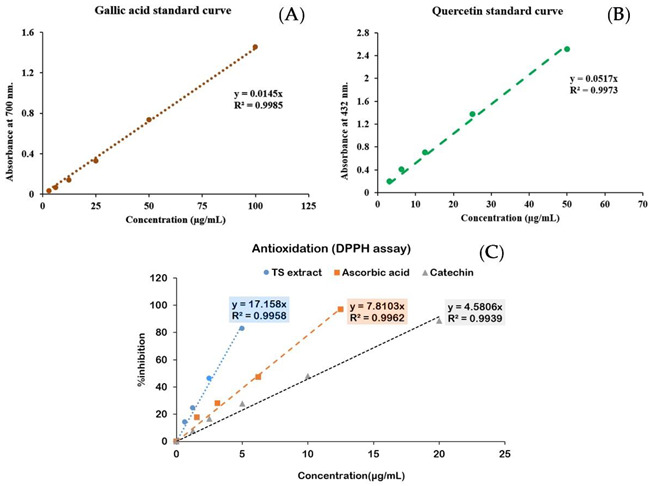

**Table 2 molecules-27-05319-t002:** Anti-microbial activities of the tamarind seed coat (TS) extract at a concentration of 250 mg/mL against *S. aureus*, *E. coli*, *P. aeruginosa*, and *C. albicans* were differently exhibited by various inhibition zone diameters. Furthermore, the determination was done compared to the controls: Dimethyl sulfoxide (DMSO, negative control); 1% Paraben (reference preservative); Ciprofloxacin (positive control).

Sample	Inhibition Zone Diameter (Mean ± sd mm; n = 2)			
	*S. aureus*	*E. coli*	*P. aeroginosa*	*C. albican*
TS extract	14.00 ± 0.64 ^b^	10.55 ± 1.56 ^b^	6.00 ± 0.00 ^b^	8.80 ± 0.28 ^b^
DMSO	Nd	Nd	Nd	Nd
1% Paraben	8.60 ± 1.34 ^b^	12.33 ± 1.73 ^b^	9.13 ± 0.95 ^b^	11.83 ± 2.16 ^ab^
Ciprofloxacin (5 µg/disc)	26.08 ± 2.02 ^a^	31.51 ± 1.68 ^a^	32.13 ± 2.95 ^a^	N/A
Nystatin (2 µg/disc)	N/A	N/A	N/A	15.85 ± 0.64 ^a^
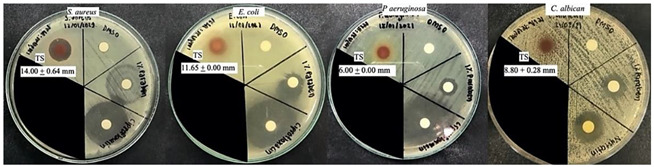				

Nd = not detectable; N/A = not applicable; ^a,b^ indicates significant difference (*p* < 0.05), compared between rows of each column.

**Table 3 molecules-27-05319-t003:** Minimal inhibitory concentration (MIC), minimal bactericidal concentration (MBC), and minimum fungicidal concentration (MFC) of the tamarind seed coat (TS) extract against *S. aureus* (A), *E. coli* (B), *P. aeruginosa* (C), and *C. albicans* (D) were defined. Interestingly, the lowest MIC and MBC to *S. aureus*; 0.03 and 3.90 mg/mL (A), manifested the potent activity of TS extract.

	*S. aureus*	*E. coli*	*P. aeruginosa*	*C. albicans*
**MIC (mg/mL)**	0.03	7.81	3.90	3.90
**MBC or MFC (mg/mL)**	3.90	31.25	15.62	31.25
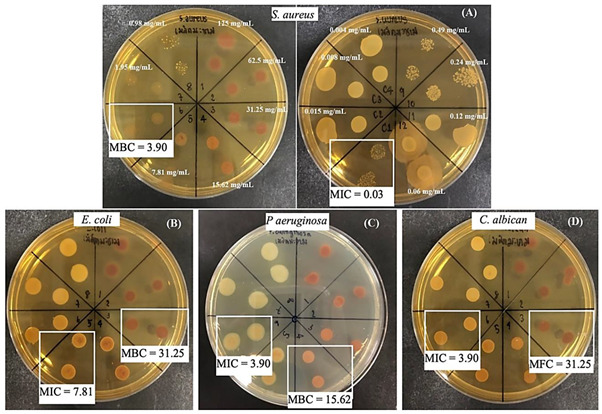				

## Data Availability

Not applicable.

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
