# Peer review of "Tamarind Seed Coat: A Catechin-Rich Source with Anti-Oxidation, Anti-Melanogenesis, Anti-Adipogenesis and Anti-Microbial Activities"

_molecules, 2022, doi:10.3390/molecules27165319_

Round 1

Reviewer 1 Report

Wandee et al studied the catechin in the seed coat of tamarind and have shown some effects including anti-oxidative, anti-microbial, and anti-melanogenesis and anti-adipogenesis activities. Although the seed coat extract is chemically simple, the bioactivity of the extract showed that it may have some potentials for further development of therapeutics. However, there are several points brought to my attention, and would like to get the authors’ further opinions. Thus I would like to suggest publication of this work after a major revision.

1.       Under table 1, there are three figures with no captions. The authors need to take care of their display and interpretation in the manuscript.

2.       Looking at fig 1A, the major peak behaves differently from catechin in many ways: 1) the same peak height did not show similar peak shapes/retentions, 2) there is a big difference in B and C for UV spectra, it is hard to convince me that they are the same compound. The authors need to show 1) co-elution of the catechin and the major peak in TS extract, in addition, the authors need to 2) show solid evidence that they are the same compound.

3.       There is another figure 1 in page 18. The authors need to take care of that.

4.       When testing the several activities, the author need to include the activities of the catechin. It does not make sense to exclude it since catechin is the most abundant peak in the TS extract based on authors’ claim.

Author Response

Thank you very much for your consideration and valuable suggestion and comments. The manuscript was reviewed carefully and revised according to your comments, for instance, section of “results,” “conclusion (L644-L645, L648-L651),” spell check, discussion, and other minor English writing (in yellow highlights). Moreover, authors’ opinion upon several points and details of revision point-by-point have been addressed in the attached file. 

Best regards,

Assoc. Prof. Khaetthareeya Sutthanut, PhD

Reviewer 2 Report

1. Title: It needs to be modified and made more apparent to the central theme, ex. extract, antioxidant, anti-cancer, and anti-microbial activities, etc.

2. M&M: It needs to be described more clearly about the plant materials, including their age, morphology, physiological conditions, etc.

3. Statistics analysis: How many replicates were conducted in each treatment for all experiments?

4. Unnecessary capital first letters were found throughout the manuscript, such as L30 Catechin (change to catechin), L31 "A"scorbic, "C"atechin, L75 "K"ojic acid, Vitamin C, Arbutin, and Deoxyarbutin—a Hydroquinone derivative, etc.

5. Table 2: Statistics should be applied to the data and provide significant differences between means. the number of replicates n=2 is too less for such kinds of experiments and it normally needs at least three replicates.

6. Table 3: Statistics should be applied to the data and provide significant differences between means.

7. The chemical constituents of the ethanolic tamarind seed coat extract should be analyzed in more detail for their quality and quantity to identify or discuss which compounds are responded to the corresponding effects.

8. L29: disc diffustion - disc diffusion

9. L59: Moreover, , - delete one comma

10. L65: physicochemical compisition - physicochemical compisition

11. L109: tamarind seed coat - the tamarind seed coat

12. L116: a HPLC analysis - an HPLC analysis

13. L176: larger diffusion area - a larger diffusion area

14. L329: unknown mechanism - an unknown mechanism

15. L625: Figure 1? It should be Figure 4.

Author Response

Thank you very much for your consideration and valuable suggestion and comments. In addition to Reviewer1’ suggestions and comments, the manuscript revision has been done according to your comments, for instance, section of “introduction,” “result,” “conclusion,” spell check, and other minor English writing (in yellow highlights). Moreover, authors’ opinion upon several points and details of revision point-by-point have been addressed in the attached file. 

Best regards,

Assoc. Prof. Khaetthareeya Sutthanut, PhD

Round 2

Reviewer 1 Report

thank the authors for their efforts in this revision, the responses are satisfactory to my questions. The authors should consider including the new data into the main figures of the manuscript. 

Author Response

Thank you very much. We are appreciated.

Best regards,

Assoc. Prof. Khaetthareeya Sutthanut, PhD

Reviewer 2 Report

The revised version is acceptable because it has been improved significantly.

Author Response

(The authors gave the same response as above.)
